# Genetics of osteopontin in patients with chronic kidney disease: The German Chronic Kidney Disease study

Yurong Cheng[1,2], Yong Li [1], Nora Scherer[1,3], Franziska Grundner-Culemann [1], Terho Lehtimäki[4,5,6], Binisha H. Mishra [4,5,6], Olli T. Raitakari[7,8,9], Matthias Nauck [10], Kai-Uwe Eckardt [11,12], Peggy Sekula [1☯], Ulla T. Schultheiss [1,13☯] *, on behalf of the GCKD investigators[¶]

**1** Institute of Genetic Epidemiology, Faculty of Medicine and Medical Center—University of Freiburg, Freiburg, Germany, **2** Faculty of Biology, University of Freiburg, Freiburg, Germany, **3** Spemann Graduate School of Biology and Medicine (SGBM), University of Freiburg, Freiburg, Germany, **4** Department of Clinical Chemistry, Faculty of Medicine and Health Technology, Tampere University, Tampere, Finland, **5** Finnish Cardiovascular Research Centre, Faculty of Medicine and Health Technology, Tampere University, Tampere, Finland, **6** Department of Clinical Chemistry, Fimlab Laboratories, Tampere, Finland, **7** Research Centre of Applied and Preventive Cardiovascular Medicine, University of Turku, Turku, Finland, **8** Department of Clinical Physiology and Nuclear Medicine, Turku University Hospital, Turku, Finland, **9** Centre for Population Health Research, University of Turku and Turku University Hospital, Turku Finland, **10** Institute of Clinical Chemistry and Laboratory Medicine, University Medicine Greifswald, Greifswald, Germany, **11** Department of Nephrology and Hypertension, University Hospital Erlangen, Friedrich-Alexander-Universität Erlangen-Nürnberg, Erlangen, Germany, **12** Department of Nephrology and Medical Intensive Care, Charité, University-Medicine, Berlin, Germany, **13** Department of Medicine IV, Nephrology and Primary Care, Faculty of Medicine and Medical Center—University of Freiburg, Freiburg, Germany

☯ These authors contributed equally to this work.
¶ a list of investigators participating in the GCKD Study can be found in S1 Information
* ulla.schultheiss@uniklinik-freiburg.de

**Data Availability Statement:** Public posting of individual level participant data is not covered by the informed patient consent form. As stated in the patient consent form and approved by the Ethics

## Abstract

Osteopontin (OPN), encoded by *SPP1*, is a phosphorylated glycoprotein predominantly synthesized in kidney tissue. Increased OPN mRNA and protein expression correlates with proteinuria, reduced creatinine clearance, and kidney fibrosis in animal models of kidney disease. But its genetic underpinnings are incompletely understood. We therefore conducted a genome-wide association study (GWAS) of OPN in a European chronic kidney disease (CKD) population. Using data from participants of the German Chronic Kidney Disease (GCKD) study (N = 4,897), a GWAS (minor allele frequency [MAF]≥1%) and aggregated variant testing (AVT, MAF<1%) of ELISA-quantified serum OPN, adjusted for age, sex, estimated glomerular filtration rate (eGFR), and urinary albumin-to-creatinine ratio (UACR) was conducted. In the project, GCKD participants had a mean age of 60 years (SD 12), median eGFR of 46 mL/min/1.73m$^2$ (p25: 37, p75: 57) and median UACR of 50 mg/g (p25: 9, p75: 383). GWAS revealed 3 loci (p<5.0E-08), two of which replicated in the population-based Young Finns Study (YFS) cohort (p<1.67E-03): rs10011284, upstream of *SPP1* encoding the OPN protein and related to OPN production, and rs4253311, mapping into *KLKB1* encoding prekallikrein (PK), which is processed to kallikrein (KAL) implicated through the kinin-kallikrein system (KKS) in blood pressure control, inflammation, blood coagulation, cancer, and cardiovascular disease. The *SPP1* gene was also identified by

Committees, a dataset containing pseudonyms can be obtained by collaborating scientists upon approval of a scientific project proposal by the steering committee of the GCKD study: (http://gckd.org). Complete summary statistics used in this project from the GCKD study can be obtained here: https://nxc-1453.imbi.uni-freiburg.de/s/gGrGW5xZRNPEHNc.

**Funding:** The work of UTS was supported within the e:Med (https://www.sys-med.de/en/) junior consortium CKDNapp (https://ckdn.app/), which is funded by grants from the German Ministry of Education and Research (BMBF, grant number 01ZX1912B; https://www.gesundheitsforschung-bmbf.de/de/ckdnapp-entwicklung-der-chronic-kidney-disease-nephrologists-app-10066.php). The work of PS was partially funded by German Research Foundation (DFG) Project-ID 431984000 - SFB 1453. The GCKD study was funded by grants from the BMBF (grant number 01ER0804) and the KfH Foundation for Preventive Medicine (https://www.kfh-stiftung-praeventivmedizin.de/content/stiftung) and corporate sponsors. Genotyping and measurements of osteopontin were supported by Bayer Pharma AG (https://www.bayer.com/en/). The Young Finns Study has been financially supported by the Academy of Finland (https://www.aka.fi/en/): grants 322098, 286284, 134309 (Eye), 126925, 121584, 124282, 129378 (Salve), 117787 (Gendi), and 41071 (Skidi); the Social Insurance Institution of Finland (https://www.kela.fi/web/en/); Competitive State Research Financing of the Expert Responsibility area of Kuopio, Tampere and Turku University Hospitals (grant X51001; https://www.vsshp.fi/en/tutkijoille/rahoitus/Pages/default.aspx); Juho Vainio Foundation (https://juhovainionsaatio.fi/en/juho-vainio-foundation/); Paavo Nurmi Foundation (https://www.paavonurmensaatio.fi/saatio_e3.htm); Finnish Foundation for Cardiovascular Research (https://www.sydantutkimussaatio.fi/en/foundation); Finnish Cultural Foundation (https://skr.fi/en); The Sigrid Juselius Foundation (https://www.sigridjuselius.fi/en/); Tampere Tuberculosis Foundation (http://www.tuberkuloosisaatio.fi/); Emil Aaltonen Foundation (https://emilaaltonen.fi/apurahat/in-english/); Yrjö Jahnsson Foundation (https://www.yjs.fi/en/); Signe and Ane Gyllenberg Foundation (https://gyllenbergs.fi/en/); Diabetes Research Foundation of Finnish Diabetes Association (https://www.diabetes.fi/en/finnish_diabetes_association/association/the_diabetes_research_foundation). This project has received funding from the European Union's Horizon 2020 research and innovation programme (https://ec.europa.eu/programmes/horizon2020/en/home) under grant agreements No 848146 (To Aition) and

AVT (p = 2.5E-8), comprising 7 splice-site and missense variants. Among others, downstream analyses revealed colocalization of the OPN association signal at *SPP1* with expression in pancreas tissue, and at *KLKB1* with various plasma proteins in *trans*, and with phenotypes (bone disorder, deep venous thrombosis) in human tissue. In summary, this GWAS of OPN levels revealed two replicated associations. The *KLKB1* locus connects the function of OPN with PK, suggestive of possible further post-translation processing of OPN. Further studies are needed to elucidate the complex role of OPN within human (patho) physiology.

## Author summary

Osteopontin (OPN) is involved in many (patho)physiological processes of the human body. Among others, it is known to be associated with adverse kidney outcomes. Since its genetic underpinnings are incompletely understood, we conducted a genome-wide association study of OPN in a European chronic kidney disease (CKD) population (N = 4,897). Of the three detected signals, two could be replicated within a population-based study of Finns. One locus is located upstream of *SPP1* which encodes the OPN protein and is related to OPN production. This gene was also disclosed by an analysis of rare variants, all presumably effecting the gene product. Another locus maps into *KLKB1* encoding prekallikrein (PK) that after processing to kallikrein (KAL) is implicated in blood pressure control and inflammation among others. Overall, our results highlight the multi-functional role of OPN and its possible pathological role in CKD. Further studies are needed to elucidate the complex role of OPN in humans.

## Introduction

Osteopontin (OPN) encoded by the *SPP1* gene was first described as a glycoprotein belonging to the SIBLING (Small Integrin-Binding LIgand N-linked Glycoprotein) family in 1985 [1]. OPN is expressed in a multitude of tissues like osteoblasts, osteocytes, odontoblasts (playing a role in mineralization and bone resorption [2,3]) macrophages, smooth muscle cells, and endothelial cells, but can also be found in the inner ear, the central nervous system, and the placenta [1,2]. Although, OPN can be detected in many cell types it is predominantly synthesized and expressed in kidney tissue. OPN production is stimulated by many factors including parathyroid hormone, calcitriol, calcium, phosphate, and cytokines. The protein is able to bind integrins through a specific peptide sequence, the arginine-glycine-aspartic acid (RGD) motif, making interaction with various cell types possible (via the nuclear factor kappa B pathway, [4,5]). In the kidney, integrins can be found in the Bowman's capsule, glomerular epithelium, and vascular epithelium [6,7]. OPN is synthesized in the thick ascending limb of Henle's loop and in the distal tubule [1,8].

In a review by Kaleta (2019), known (patho)physiological roles of OPN have been discussed [1]. Based on this review, the physiological role of OPN in the kidney is not fully understood yet, but it has been suggested as being essential for tubulogenesis [1]. *SPP1* mRNA as well as OPN protein expression were elevated in mostly rat models of kidney diseases and high OPN expression correlated with proteinuria, reduced kidney function, and fibrosis [1]. One study identified various polymorphisms in the *SPP1* promoter region affecting its transcriptional activity [9]. In the past several specific *SPP1* gene variants have been associated with the

No 755320 (TAXINOMISIS). This project has received funding from the European Research Council (ERC; https://erc.europa.eu/) advanced grants under grant agreement No 742927 (MULTIEPIGEN project); Tampere University Hospital Supporting Foundation (https://www.tays.fi/en-US/Research_and_development) and Finnish Society of Clinical Chemistry (https://www.ifcc.org/). The funders had no role in study design, data collection and analysis, decision to publish, or preparation of the manuscript. We acknowledge support by the Open Access Publication Fund of the University of Freiburg.

**Competing interests:** The authors have declared that no competing interests exist.

pathogenesis and progression of different kidney diseases. Other case-control studies reported on specific variants in the *SPP1* gene being associated with different kidney disease patients in comparison to a (healthy) control group: For example, rs1126616 was repeatedly reported as a marker for lupus nephritis and immunoglobulin A nephropathy [10–14]. In connection with diabetic nephropathy, the two SNPs in *SPP1*, rs11730582 and rs17524488, have been reported [15,16]. We therefore reasoned that the presence of reduced kidney function may represent a good study setting to further establish our understanding of the genetic underpinnings of OPN levels in kidney disease, as some biologic mechanisms might be upregulated and thus be easier to detect, which has been shown before [17–19]. In Jing et al. [19], for example, the magnitude of effects for known loci identified in a GWAS of serum urate in CKD patients were of similar or higher magnitude than those reported from population-based studies.

The German Chronic Kidney Disease (GCKD) study comprises a large cohort of CKD patients [20]. Besides demographic and clinical data, genetic data are available as well as baseline measurements of serum OPN, providing an ideal setting to explore the genetics of OPN. For this purpose, we performed a GWAS of serum OPN levels in the GCKD study.

## Results

### Description of the GCKD analysis set

Table 1 gives an overview of baseline characteristics of a selected number of variables for the complete GCKD study cohort and the GWAS analysis set in which participants with complete data on genetics, OPN measurements as well as estimated glomerular filtration rate (eGFR) and urinary albumin-to-creatinine ratio (UACR) are included (S1 Fig). There were no major discrepancies between the complete cohort and the analysis set.

Overall, the GWAS analysis set was characterized by a proportion of 60% men with a mean age of 60.2 years (SD: 12.0), with median values of 46.0 mL/min/1.73m$^2$ (p25: 37.0; p75: 57.0) for eGFR and of 50.2 mg/g (p25: 9.4; p75: 382.8) for UACR (Table 1). Among the included

**Table 1. Study sample characteristics of the complete GCKD study cohort (N = 5,217) and the GWAS analysis set (N = 4,897).**

|  | N = 5,217 | N = 4,897 |
|---|---|---|
| **Osteopontin, ng/mL, median (p25; p75)** | 29.2 (20.7; 41.9) | 29.2 (20.7; 41.8) |
| **Age, years, mean (SD)** | 60.1 (12.0) | 60.2 (12.0) |
| **Male, N (%)** | 3,132 (60.0) | 2,950 (60.2) |
| **eGFR, mL/min/1.73m$^2$, median (p25; p75)** | 46.4 (37.1;57.4) | 46.0 (37.0; 57.0) |
| **UACR, mg/g, median (p25; p75)** | 50.9 (9.7; 391.7) | 50.2 (9.4; 382.8) |
| **HDL, mg/dL, median (p25; p75)** | 48.4 (39.3; 61.4) | 48.5 (39.4; 61.4) |
| **Systolic blood pressure, mmHg, mean (SD)** | 139.5 (20.4) | 139.4 (20.3) |
| **BMI, kg/m$^2$, mean (SD)** | 29.8 (6.0) | 29.8 (6.0) |
| **Diabetes mellitus, N (%)** | 1,868 (35.8) | 1,715 (35.0) |
| **Smoking, current, N (%)** | 828 (15.9) | 781 (16.0) |
| **CVD, N (%)** | 1,591 (30.5) | 1,489 (30.4) |

Continuous variables are mean (SD: standard deviation) for normally distributed variables or median (p25; p75: 25$^{th}$; 75$^{th}$ percentile) for variables with skewed distributions.

eGFR: estimated glomerular filtration rate; UACR: urinary albumin to creatinine ratio; HDL: high density lipoproptein; BMI: body mass index; CVD: history of cardiovascular disease.

*Missingness per variable*: N complete cohort (N GWAS cohort): Osteopontin 63 (0), eGFR 55 (0), UACR 90 (0), HDL 66 (8), systolic blood pressure 34 (29), BMI 54 (49), smoking 16 (14), CVD 2 (2).

participants, 35% had a prevalent diabetes mellitus, 16% were current smokers and 30% reported a history of cardiovascular disease (CVD).

Median OPN levels in the complete cohort were 29.2 ng/mL (p25: 20.7; p75: 41.9; **Table 1**). Levels of OPN increased on average across eGFR categories and UACR categories from a median of 25.4 ng/mL for CKD stage G1/2 to 38.5 ng/mL for CKD stage G4/5, as well as from median OPN values of 25.6 ng/mL for UACR stage A1 to mean OPN values of 34.2 ng/mL for UACR stage A3 (**S2 Fig**).

### Genome-wide association study and fine-mapping

We conducted a GWAS for serum OPN levels ($log_2$-transformed) using ~7.7 million high-quality autosomal bi-allelic variants of the GCKD study with a minor allele frequency (MAF) of $\geq 0.01$ (**S1 Table**). The quantile-quantile plot comparing observed and expected p-values from the OPN GWAS did not indicate inflation (inflation factor $\lambda = 1.01$), consistent with the absence of systematic errors (**S3 Fig**).

Overall, the Manhattan plot revealed three genome-wide significant regions associated with OPN levels (p-value <5.0E-08; **S4 Fig**). Besides the three identified regions, conditional analysis did not reveal additional independent signals (**S5 Fig**). The respective association results for the three index SNPs (= SNP with the lowest p-value in the respective region) are presented in **Table 2**. For all three SNPs the coded allele was present frequently (allele frequency range 0.5–0.75). The respective coded allele in our cohort decreased OPN levels on average (**S6 Fig**) with effect estimates per copy of the coded allele ranging from -0.10 to -0.18 (SE: 0.01–0.02; **Table 2**). One of the index SNPs on chromosome 4 (rs10011284, 4:88833389) is located upstream of *SPP1*, which encodes the protein OPN itself (**Fig 1A**). The other index SNP on chromosome 4 (rs4253311, 4:187174683) maps into the *KLKB1* gene (intronic variant), encoding the protein prekallikrein (PK) that is converted to kallikrein (KAL); a protease implicated in the surface-dependent activation of coagulation, bradykinin (BK) release, and potentially the renin angiotensin aldosterone system (**Fig 2A**). The index SNP on chromosome 5 (rs2731673, 5:176839898) maps closest to the *F12* gene, which encodes coagulation factor XII, a serine protease that cleaves *KLKB1*-encoded PK to KAL, among other functions and is also related to blood coagulation, fibrinolysis, and the generation of BK (**S7 Fig**). A summary of

**Table 2. Association results for the 3 index SNPs genome-wide-significantly associated with serum osteopontin levels in the GWAS discovery of the GCKD study (N = 4,897) and in the replication cohort of the YFS (N = 1,979).**

| SNP | Position (GRCh37) | Gene(s) | Coded allele / non-coded allele | Study | Quality (quality score) | Frequency, coded allele | Beta (SE) | p-value, 2-sided |
|---|---|---|---|---|---|---|---|---|
| rs10011284 | 4:88833389 | *MEPE* (dist = 65421), *SPP1* (dist = 63413) | A/G | GCKD | imputed (0.999) | 0.57 | -0.10 (0.01) | **8.59E-11** |
| | | | | YFS | imputed (0.997) | 0.52 | -0.07 (0.02) | **2.11E-05** |
| rs4253311 | 4:187174683 | *KLKB1* | G/A | GCKD | genotyped | 0.50 | -0.14 (0.01) | **5.29E-20** |
| | | | | YFS | imputed (0.997) | 0.58 | -0.10 (0.02) | **1.93E-08** |
| rs2731673 | 5:176839898 | *F12* (dist = 3321), *GRK6* (dist = 13789) | C/T | GCKD | imputed (0.987) | 0.75 | -0.18 (0.02) | **4.47E-25** |
| | | | | YFS | imputed (0.985) | 0.74 | -0.03 (0.02) | 1.63E-01 |

Associations with OPN were adjusted for age, sex, log(eGFR), log(UACR) in GCKD (GWAS discovery) and for age, sex, and $log_2$(eGFR) in YFS (replication cohort).
Statistical significant association p-values are marked in bold: discovery: p-value<5E-08, replication: 1-sided p-value<0.05/3.

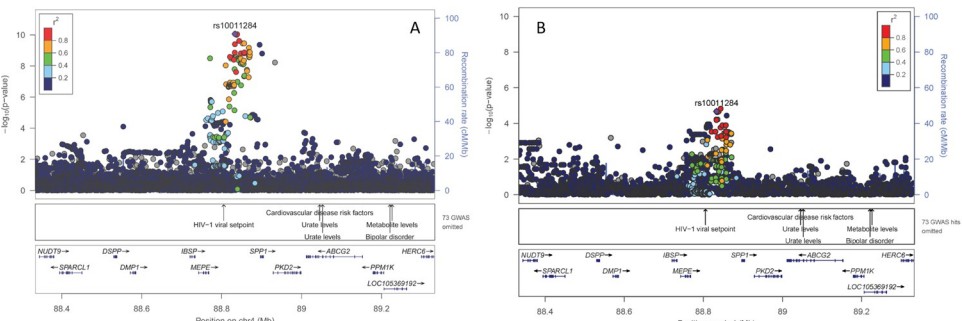

**Fig 1. Regional association plot for the region around rs10011284 on chromosome 4: (A) GCKD study (discovery) and (B) YFS cohort (replication).** Plots are produced in LocusZoom and show the most strongly associated SNP (purple diamond), SNP colors reflect LD correlation ($r^2$) using 1000G EUR population as reference. The -$\log_{10}$ p-values (left y-axis) of SNPs are shown according to their chromosomal positions (x-axis, GRCh37); the genetic recombination rates are shown on the right y-axis. The -$\log_{10}$ p-values are shown for both genotyped and imputed SNPs distributed in a 0.8-megabase genomic region.

annotations combined from different publicly accessible data bases and related to all three SNPs is provided in **S2 Table**.

We next tested whether these three index SNPs were associated with serum OPN levels in the Young Finns Study (YFS) cohort, a population-based study with a mean age of 38 years (SD: 5.0) and a mean eGFR of 92.6 mL/min/1.73m$^2$ (SD: 20.7; **S1 Methods**). Both SNPs on chromosome 4, rs10011284 and rs4253311, were significantly associated with OPN levels showing also direction consistency (**Table 2 and Figs 1B and 2B**). In contrast, rs2731673 closest to the *F12* gene did not replicate in the YFS cohort (**S7 Fig**).

The two replicated SNPs on chromosome 4 explained 1% of OPN levels each within the GCKD study and did not show a non-additive effect on OPN levels (**S8 Fig**). Moreover, statistical fine-mapping was performed for the two replicated loci to resolve associated loci into potentially causal variants by constructing credible sets that collectively accounted for 99% posterior probability of containing the variant or variants that cause the association signal (PPA; **Material and methods,** [21]). However, fine-mapping results are inconclusive as both constructed sets are large and single variants included only exhibit low PPA estimates (**S3 Table**).

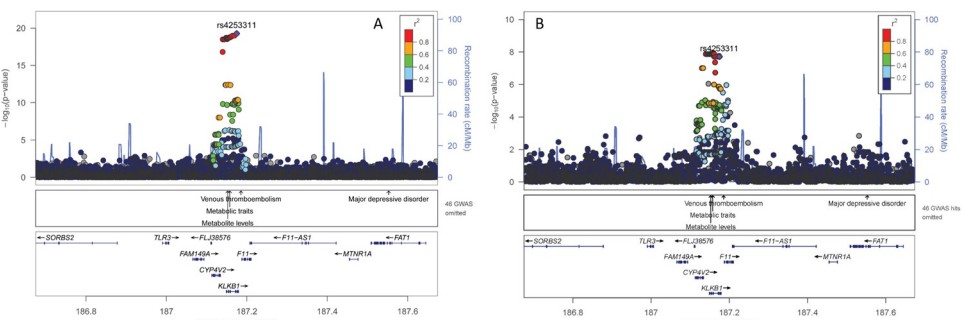

**Fig 2. Regional association plot for the region around rs4253311 on chromosome 4: (A) GCKD study (discovery) and (B) YFS cohort (replication).** Plots are produced in LocusZoom and show the most strongly associated SNP (purple diamond), SNP colors reflect LD correlation ($r^2$) using 1000G EUR population as reference. The -$\log_{10}$ p-values (left y-axis) of SNPs are shown according to their chromosomal positions (x-axis, GRCh37); the genetic recombination rates are shown on the right y-axis. The -$\log_{10}$ p-values are shown for both genotyped and imputed SNPs distributed in a 0.8-megabase genomic region.

## Colocalization analyses

In order to learn more about the molecular mechanisms and associated phenotypes underlying the identified association signals for OPN, we compared patterns of OPN GWAS results in predefined regions to respective GWAS summary statistics from three other sources using human data (see **Material and methods** for details). Comparable patterns may indicate a common biological basis.

**Gene expression.** First, we performed colocalization analyses of OPN GWAS summary statistics related to the two replicated loci with the corresponding GWAS summary statistics of gene expression in *cis* using data from the GTEx project and the NEPTUNE study (**Material and methods**). Colocalization (posterior probability of H4 [p12] >0.8) of the OPN association signals were detected with the expression of *MEPE* in lung (**Fig 3A**) and of *SPP1* in pancreas (**Fig 3B** and **S4 Table**). Furthermore, colocalization was found between the OPN association signal and expression of *F11* in six other tissues (in descending order of H4): tibial artery (**Fig 3C**), brain cortex, terminal ileum part of the small intestine, muscularis of the esophagus, transverse colon, and aortic artery (**S4 Table**). Except for the colocalization of the OPN signal with expression of *MEPE* in lung, the effect direction of both traits (OPN and gene expression) was concordant (alpha12>0; **S4 Table**).

**Plasma proteome.** In addition, colocalization analyses were conducted using GWAS summary statistics of SNP associations in *cis* and in *trans* with levels of ~3,000 different plasma proteins (pGWAS) reported by Sun *et al.* (**Material and methods,** [22]). While no colocalization was present for the summary statistics of the GWAS of OPN and proteins in *cis*, pGWAS results for 87 proteins from various protein classes were found to *trans* colocalize with OPN GWAS results for rs4253311 at *KLKB1* (**S5 Table**). For the majority of colocalization results

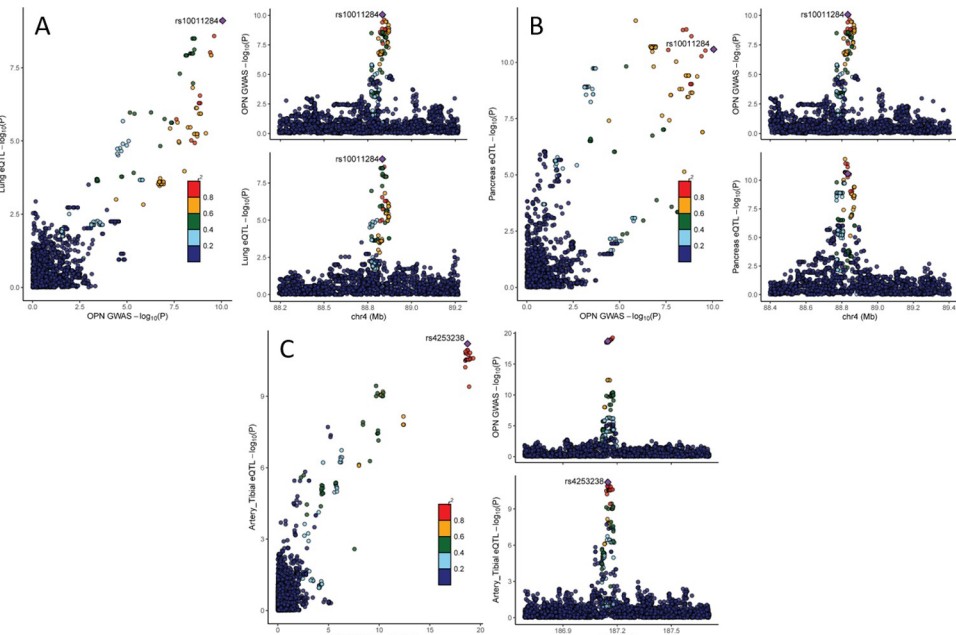

**Fig 3. Comparing summary statistics from OPN GWAS and GTEx tissue colocalizing: (A)** *MEPE*: **OPN and lung tissue (H4: p12 = 0.99), (B)** *SPP1*: **OPN and pancreas (H4: p12 = 0.85); (C)** *F11*: **OPN and tibial artery (H4: P12 = 0.98).** Left: scatter plot comparing association p-values from both sources against each other (-log10 scale). Upper right: OPN GWAS results for the region of interest. Lower right: GTEx GWAS results for the region of interest of respective organ. Colors reflect LD correlation ($r^2$) using 1000G EUR population as reference.

(60/87, 69%), the effect direction of OPN and proteins was concordant, which is in accordance with the fact that the activated KKS is involved in a broad spectrum of processes, like inflammation, cancer, cardiovascular disease, as well as (patho)physiological roles in kidney and the central nervous system.

For 86 mapped proteins, a Gene Ontology (GO) enrichment analysis was conducted to assess whether their encoding genes were enriched in terms representing specific cellular components and molecular functions (**Material and methods**, **S6 Table**). Because of a hierarchical structure of reference lists, implicated terms are partially dependent on each other showing e.g. overlapping upregulated molecular functions as well as upregulated hormone and receptor activities (**S9 Fig**).

**UK Biobank (UKB) diseases.**   In order to address the interplay between genetic modulation of OPN levels and phenotypes we further conducted a colocalization analysis with binary UKB disease traits that showed a genome-wide significant association in the GeneAtlas resource [23]. Based on marginal association statistics, no positive colocalization was detected between OPN and various disease traits (**S7 Table**). Still, for four of seven traits with an association signal different from the OPN signal in the region (H3: p1.2>0.8), a second independent association signal for the respective disease trait in UKB was detected. We then performed additional colocalization analyses based on the obtained conditional association statistics, and identified a positive colocalization between the OPN association signal at rs4253311 at the *KLKB1* locus and rs1593 for deep venous thrombosis (DVT; H4: p12 = 0.96; **S7 Table** and **S10 Fig**). In the GeneAtlas GWAS of DVT, the major allele of rs1593 (A, allele frequency = 0.87) was associated with a higher risk for DVT (OR = 1.2, p-value 2.4e-16), as was the major allele of rs4253311 (major allele: G, allele frequency = 0.51, OR = 1.13, p-value = 2.6e-16).

## Rare variant analysis

Based on 4,879 GCKD participants with available exome chip data, we additionally conducted aggregated rare variant testing (**S1 Fig**). Variants with a MAF <1% and having a major effect on the gene product (nonsynonymous, stop gain/loss, splicing; "qualifying variants") as annotated by dbNSFP v.2.0 were aggregated (**Material and methods**, [24,25]).

While there was no significant association result when the burden test was used, we found a significant association using the sequence kernel association test (SKAT) for the *SPP1* gene (**Table 3**, p-value = 2.5E-08), which remained significant after adjustment for the two replicated SNPs from the initial GWAS (p-value = 9.4E-08). Seven variants were aggregated for the analysis of the *SPP1* gene. In the single variant analysis, effect estimates of the seven variants ranged from -1.20 to 0.24 with rs139555315 (4,88901197), a splice-site variant (CADD score: 23.7) showing the most significant association (effect estimate = -1.20, SE = 0.19, p-value = 2.5E-10) and thus likely driving the association. This is supported by the non-significant result (p-value$_{SKAT}$ = 3.5E-01) when rs139555315 was excluded from the variant set.

Other genes implicated by GWAS of common variants reached only nominal statistical significance (p<0.05; *MEPE*: p-value$_{SKAT}$ = 4.6E-03; *KLKB1*: p-value$_{Burden}$ = 5.6E-04, p-value$_{SKAT}$ = 2.8E-03; *F12*: p-value$_{Burden}$ = 3.4E-02, p-value$_{SKAT}$ = 3.0E-02).

## Discussion

In this study, we focused on characterizing the genetics of OPN within a CKD cohort, because OPN levels are known to be associated with adverse kidney outcomes, but genetic underpinnings of this kidney-enhanced protein are not fully understood. The use of a CKD patient cohort might present an advantageous setting in which the transcription of kidney-specific genes may be altered in comparison to the general population, making identification of

**Table 3. Association results for *SPP1* (chromosome 4) for the rare variant analysis.**

**(A) Association results of aggregated variant testing**

| Model | Gene | Burden test | | SKAT test | No of SNPs aggregated | Total MAC | Cumulative MAF of SNPs |
|---|---|---|---|---|---|---|---|
| | | p-value | Beta (SE) | p-value | | | |
| adjusted for age, sex, log(eGFR), log (UACR) | *SPP1* | 1.36E-03 | -0.31 (0.10) | **2.51E-08** | 7 | 59.03 | 6.05E-03 |
| Same as above plus 2 replicated common SNPs* | *SPP1* | 1.50E-03 | -0.30 (0.10) | **9.43E-08** | 7 | 59.03 | 6.05E-03 |

**(B) Association results of single variant analysis for aggregated variants**

| Model | Variant | Position (GRCh37) | MAF | Beta (SE) | p-value | Exonic effect (CADD) |
|---|---|---|---|---|---|---|
| adjusted for age, sex, log(eGFR), log (UACR) | rs139555315 | 88901197 | 1.54E-03 | -1.20 (0.19) | **2.54E-10** | splicing (23.7) |
| | rs140258871 | 88901249 | 1.95E-03 | 0.24 (0.17) | 1.62E-01 | nonsynonymous (NA) |
| | rs138638879 | 88902774 | 7.18E-04 | -0.18 (0.28) | 5.18E-01 | nonsynonymous (26.8) |
| | rs7435825 | 88903774 | 3.08E-04 | 0.04 (0.42) | 9.17E-01 | nonsynonymous (12.8) |
| | rs149833253 | 88903825 | 3.08E-04 | -0.45 (0.42) | 2.85E-01 | nonsynonymous (8.8) |
| | rs146563765 | 88903899 | 1.02E-03 | -0.10 (0.23) | 6.83E-01 | nonsynonymous (4.2) |
| | rs4660 | 88904005 | 2.05E-04 | -0.52 (0.52) | 3.22E-01 | nonsynonymous (9.8) |

MAC: minor allele count; MAF: minor allele frequency; NA: not available

* rs10011284, rs4253311. Statistical significant association p-values are marked in bold (aggregated variant testing: p-value<1.4E-06, single variant analysis: p-value<5.0E-08. Exonic effects: source dbNSFP v.2.0.

specific SNPs possibly easier. We identified three loci, two on chromosome 4 (rs10011284, rs4253311), that could be replicated in an external population-based cohort, and rs2731673 on chromosome 5, which could not be replicated. When aggregating rare variants, the *SPP1* gene encoding OPN was detected.

To our knowledge this is the first GWAS of serum OPN levels quantified via ELISA. Other studies like Sun *et al.* conducted GWAS of plasma proteins (pGWAS) including OPN. Here proteins were quantified differently via an aptamer-based technology (SOMAscan, [22]). While this study was likely too small (N = 3,301) to detect any genetic signals for OPN, a study by Pietzner *et al.* (N up to 10,708) provided signals on chromosome 3, 4, and 10 associated with SOMAscan-measured OPN [26]. The detected signal on chromosome 4 (rs5860110, 4:88897106, MAF = 0.30) is a common, intronic indel located within *SPP1* not in linkage disequilibrium (LD) with common variants identified in our project. In a next step, Pietzner *et al.* reported association statistics for the SOMAscan detected loci using a different technique to measure proteins (Olink, antibody-based protein panels). Olink measurements were, however, only available for a small fraction of the study population and none of the three SOMAscan-measured OPN signals could be confirmed. The systematic comparison of protein levels quantified by these two techniques revealed varying correlations (median 0.38, IQR: 0.08–0.64). For OPN, a correlation coefficient of 0.51 was reported. A similar comparison of even more proteomics platforms also reported on a wide range of correlations among measurements [27]. Differences in the detection of genetic signals could thus not only be explained by differences in power but by technical, protein and variant characteristics. Across platforms, any comparison results is thus difficult and meta-analyses could lead to wrong inferences.

### rs10011284; 4:88833389 (*SPP1/MEPE* intergenic)

The index SNP rs10011284 (MAF = 0.43) of the first locus on chromosome 4 maps between the *SPP1* and *MEPE* genes. Whether this variant or another is the causal variant underlying the observed association remains unclear as results from statistical fine-mapping are inconclusive. Nevertheless, when aggregating rare variants, *SPP1*, the gene encoding the protein OPN, showed a significant association driven by a splice-site variant. Any errors occuring during the splicing process can lead to false intron removal causing alterations of the open reading frame. In turn this may either lead to formation of a premature stop codon and a shortened protein or more likely to a faster mRNA degradation called nonsense mediated decay [28]. The variant driving this association at *SPP1*, a splice-site variant (rs139555315, 4:88901197, MAF = 1.54E-03), was reported to be associated with pediatric systemic lupus erythematosus [29].

*SPP1* is made up of 7 exons containing 942 transcribed nucleotides from the start codon in exon 2 to the stop codon (within exon 7, [30]). OPN belongs to the SIBLING glycoprotein family of secreted phosphoproteins; other members of this family comprise dentin matrix protein 1, dentin-sialophosphoprotein, statherin, bone sialoprotein, and matrix extracellular phosphoglycoprotein (MEPE). Results from our colocalization analysis using GeneAtlas are pointing towards a connection of *SPP1* with bone disorders. Fitting with these results, one of OPN's main physiological functions in the body is the regulation of biomineralization processes [31]. Conflicting results have been reported on OPN as well as *SPP1* polymorphisms and susceptibility to nephrolithiasis in the past [32–36]. From in vitro studies it may be inferred, that OPN inhibits nucleation, growth, and aggregation of calcium oxalate crystals [37], but clinical studies draw a more unclear picture. Some researchers report on a protective role of OPN, where others do not [38,39]. Nonetheless, a recent study from South Asia found a significant association of the *SPP1* rs2853744:G>T polymorphism with urolithiasis [40].

OPN is mostly secreted, but an intracellular form has also been reported [41]. Using reverse-transcription-PCR OPN was found to be expressed in normal human adult kidney, further immunohistochemical analyses and in situ hybridization revealed OPN expression to be restricted to the distal convoluted and straight tubules in kidney cortex and medulla in monkey kidney [42]. Looking at GTEx tissue expression data, a positive colocalization of the OPN association signal at *SPP1* with pancreas tissue was detected. This is in line with findings in the literature where OPN has been suggested to have a role in type 2 diabetes. One study performed by Cai *et al.* investigated a diabetic mouse model SUR1-E1506K+/+ and islets from human donors and was able to demonstrate that in islets from human cadaver donors, *OPN* gene expression was elevated in diabetic islets, and externally added OPN significantly increased glucose-stimulated insulin secretion from diabetic but not normal glycemic donors [43]. Many other studies have also investigated OPN's role in pancreatic cancer, here, OPN was found to be a prognostic marker associating higher levels with poor overall prognosis in patients [44].

*MEPE* (Matric Extracellular PhosphoglycoprotEin) is the gene encoding the secreted calcium-binding phosphoprotein MEPE. A common feature of SIBLING proteins is the Acidic Serine Aspartate Rich MEPE associated motif (ASARM), involved in the regulation of mineralization, bone turnover, mechanotransduction, phosphate and energy metabolism [45]. The ASARM motif is also the connecting link between SIBLINGs and FGF23 thereby being part of the physiological bone-kidney link [45]. MEPE is involved in the regulation of the phosphate homeostasis controlled by the kidney and intestine [46,47]. Colocalization of the OPN association signal at *MEPE* leads to detection of an association with lung tissue. So far a connection between several members of the SIBLING family with lung cancer have been reported, but a definite connection between *MEPE* and lung has not been made before [48]. Since there are

multiple transcript variants known due to alternative splicing a connection between *MEPE* and lung cannot be ruled out and could offer possible research areas in the future. Results from our colocalization analysis using GeneAtlas are pointing towards a connection of *MEPE* with bone disorder. Diseases associated with MEPE are osteomalacia and autosomal-dominant hypophosphatemic rickets supporting this connection between *MEPE* and bone disorders [49]. Another phenotype seemingly related to rs10011284 is gout. This association is most likely driven by *ABCG2*, which is located in close proximity to *SPP1* and *MEPE*, and is one of the best described uric acid transporter genes to date [19,50].

### rs4253311; 4:187174683 (*KLKB1* intronic)

The 2nd replicated index SNP rs4253311 (MAF = 0.50) on chromosome 4 is an intronic variant of the *KLKB1* gene. Again, whether this variant is the causal variant responsible for the observed association cannot be answered by this study.

*KLKB1* encodes prekallikrein (PK) a single-chain zymogen that, after activation to kallikrein (KAL), a serine protease, becomes involved in the surface-dependent activation of blood coagulation, fibrinolysis, kinin generation and inflammation. Diseases associated with *KLKB1* include PK deficiency and malignant essential hypertension [51,52]. PK and subsequently KAL are part of the kallikrein-kinin system (KKS). Main function of KAL includes the release of bradykinin (BK) [53]. Genome-wide association studies in the past identified associated SNPs in *KLKB1* with vasoactive peptides or precursors of vasoactive peptides (BK [54,55], active renin [56], B-type natriuretic peptide [57], aldosterone/renin ratio [57], midregional proadrenomedullin and C-terminal-pro-endothelin-1 [58], L-arginine [59]), and apolipoprotein A IV [60], but not OPN.

Pathways related to this gene include complement and coagulation cascades, as well as degradation of the extracellular matrix [61,62]. An important paralog of *KLKB1* is the gene *F11*. Since our analysis revealed colocalizations between OPN association signal for rs4253311 and expression in multiple tissues for *F11*, it could be presumed that rs4253311 is linked to *F11* rather than *KLKB1*. Interestingly, OPN contains several protease cleavage sites that regulate its activity [31]. Some OPN interaction sites require cleavage by thrombin, another serine protease, to become fully functional. In return, OPN has been shown to be a substrate for other proteases, that regulate its activity [31]. One might speculate that inflammatory processes within the kidney of CKD patients bring together, on the one hand, an activated KKS and, on the other hand, higher OPN levels, thus it might be plausible that new bioactive OPN fragments could possibly be generated by KAL.

Conditional colocalization analyses with SNPs located around *KLKB1* resulted in positive results for rs1593, mapping intronically into the *F11* gene, and DVT. DVT is a serious disease influenced by both genetic and environmental risk factors, but 60% of the variation in risk for DVT has been attributed to genetic risk factors in the past [63]. Genetic studies of DVT have reported several common SNPs in the 4q35.2 locus to be associated with DVT [63]. These common SNPs were localized within *KLKB1* and *F11* amongst others [63].

Other colocalization results showed association signals between the OPN locus at *KLKB1* and 87 plasma proteins. These proteins showed enrichment for proteins of the neuronal cell body in plasma and cerebrospinal fluid of patients. BK, the principal effector of the plasma KKS, is generated systemically and locally (vessel wall) and acts in a paracrine or autocrine way influencing vascular tone and ultrastructure via two G protein-coupled receptors [64,65]. Components of the KKS and in particular BK have been shown to have important functions in the central nervous system by regulating cerebrovascular resistance, vessel capacity and permeability of the blood-brain-barrier. Maintenance of a vascular permeability equilibrium in the

central nervous system is critical for maintaining brain integrity. Associations of the KKS in CNS pathology include several disease states among which are neuropsychiatric lupus, Alzheimer's disease, schizophrenia, and epileptic syndrome [66]. These facts in turn explain the enrichment analysis results of the molecular functions: hormone activity via signaling receptor binding as well as receptor regulatory activity.

### rs2731673; 5:176839898 (*F12/GRK6* intergenic)

Finally, the index SNP rs2731673 (MAF = 0.25) that could not be replicated in the population-based YFS cohort is located between the genes *F12* and *GRK6*. While this non-confirmation may indicate a false positive result of our GWAS, replication may have failed for some unknown reason such as being a specific result relevant for CKD patients. In the absence of another cohort (whether population-based or CKD cohort) with necessary data on OPN and genetics, we were unable to validate this any further.

The gene nearest to the locus is the *F12* gene that encodes coagulation factor XII, which, together with plasma PK, belongs to the contact activation system [67,68]. While KAL can activate factor XII (factor XIIa: active enzyme of factor XII) that, in turn promotes inflammation via the KKS, including PK [69]. Since CKD patients markedly have more inflammation and fibrosis as the joined common final path of kidney disease progression these insights and connection may encourage further validation of this locus. Inflammatory processes also play a major role in CVD and CKD patients, who are well known to suffer from excessive CVD promoting higher morbidity and mortality. In the human cardiovascular system, OPN is primarily expressed in endothelial cells, macrophages, and smooth muscle derived foam cells and can also be detected in human atherosclerotic plaques of the arterial system [70,71]. Higher serum OPN levels were found in patients with acute coronary syndrome vs chronic coronary syndrome. In coronary artery disease (CAD) patients high OPN levels were associated with rapid coronary plaque progression and in-stent restenosis [72]. OPN has been known to be associated with adverse outcomes in patients with CVD [73–75], but its function in CVD is diverse. Acute increases of OPN in CVD are associated with wound healing and neovascularization [76,77]. Chronically increased OPN, however, is associated with a poor prognosis of major adverse cardiovascular events [78].

*GRK6* on the other hand is involved in blood pressure regulation and was found to have a decreased kidney expression in spontaneously hypertensive rats, providing a similar rationale [79].

### Strengths and limitations

The presented analyses have several strengths and limitations: Firstly, our analyses are based on a CKD patient population of European ancestry and mostly CKD stage 3 under regular nephrologist care. While biological mechanisms may be upregulated in impaired kidney function and thus detected more easily in CKD patients, results potentially compromise generalizability to the general population as well as to other ethnicities. Secondly, we could replicate two of three identified loci in a population-based cohort of the YFS, who also applied an ELISA technique to measure OPN, thus confirming the potential to transfer findings from a CKD cohort to the general population. Regarding the third, non-replicated finding, one should await further validation of this result as regarding it as false-positive would be a premature conclusion. Thirdly, serum OPN was measured in the GCKD study from baseline samples using a state-of-the-art ELISA assay. Although serum has been validated for use in the used OPN assay, it is not the recommended sample type, because of proteolytic cleavage by thrombin during the clotting process. In contrast, OPN measurements were obtained from plasma

samples in the YFS cohort. While a comparable assay was used, differences between levels in CKD patients and YFS participants might thus be explainable by more than disease status.

## Conclusions

In this first GWAS of serum OPN levels in a large CKD cohort two replicated associations on chromosome 4 were detected. One locus closest to the *SPP1* gene, as well as a locus mapping into the *KLKB1* gene, connecting OPN to its production and the KKS. Further studies are needed to fully explain OPN's role in kidney (patho)physiology and elucidate functions of OPN in connection with the KKS and possibly inflammatory processes during kidney fibrosis.

## Material and methods

### Ethics statement

The German Chronic Kidney Disease (GCKD) study was approved by all Ethics Committees of participating institutions in Germany that also covers the present project. It was registered in the national registry for clinical studies (DRKS 00003971; **S1 Information**). Written informed consent was obtained for all participants.

### Study population

The GCKD study cohort consists of 5,217 adult CKD patients of European ancestry with (i) an eGFR between 30–60 mL/min per 1.73m$^2$ or (ii) an eGFR >60 mL/min/1.73 m$^2$ and 'overt' albuminuria/proteinuria at baseline [20]. At the baseline visit (2010–2012), trained personnel obtained data using a standardized questionnaire and physical examinations. Biosamples were obtained, directly processed and then stored at -80˚C in a central biobank [80]. Study procedures and main baseline findings have been reported before [20,81].

### Baseline variables and measurements

A standardized set of biomarkers was measured in a central certified laboratory using standardized protocols [81]. Among others, creatinine and albumin from serum and urine were quantified using an IDMS traceable methodology (Creatinine plus, Roche, Germany) and a turbidimetric method (Tina-quant, Roche, Germany) Roche/Hitachi MODULAR P, respectively.

Glomerular filtration rate (GFR) was estimated using the creatinine-based CKD-EPI formula (unit: mL/min/1.73m$^2$, [82]). UACR was calculated as measured urinary albumin/urinary creatinine (mg/g, [83]). Age and sex were self-reported at the baseline visit.

In 2015, OPN was measured from baseline serum samples of the complete GCKD study cohort using a quantitative sandwich enzyme immunoassay technique (solid-phase ELISA; Quantikine Human OPN Immunoassay DOST00 from R&D Systems (R&D Systems Europe, Abingdon, UK)). Quantification was carried out at the Institute of Clinical Chemistry and Laboratory Medicine, Greifswald, Germany. Coefficients of variation (intra-assay) were 4.5%, 5.3% and 3.5% for low, median and high levels, respectively. The inter-assay coefficient of variation was 6.4%. Reagents and secondary standards were used as recommended by the manufacturer.

### Genotyping, quality control and imputation

Detailed information on genotyping and data cleaning in the GCKD study has been described previously [18]. Briefly, DNA was isolated from whole blood and genotyped at 2,612,357 variants for 5,123 GCKD participants using the Illumina HumanOmni2.5 Exome BeadChip array

(Illumina, GenomeStudio, Genotyping Module Version 1.9.4) at the Helmholtz Center Munich. Data cleaning was carried out separately for the Omni2.5 content and the exome chip content of the array.

Based on standardized protocols [84], custom written scripts (R, Perl) and Plink1.9 [85] software was used for quality control (QC) of the Omni2.5 content. Sample-based QC steps included checks of call rate, sex, heterozygosity, genetic ancestry and relatedness, leading to the exclusion of 89 samples. On the variant level, single nucleotide polymorphisms (SNPs) were excluded if the call rate was <0.96, and whenever the assumption of the Hardy-Weinberg equilibrium was violated (p-value <1.0E-05). After removing SNPs on duplicate positions, the cleaned dataset contained 5,034 individuals and 2,337,794 SNPs (**S1 Fig**). Genotypes were then imputed using minimac3 v2.0.1 at the Michigan Imputation Server [86]. The Haplotype Reference Consortium (HRC) haplotypes version r1.1 were used as the reference panel, and Eagle 2.3 was used for phasing. The final dataset contains data of 5,034 participants and 7,750,367 high-quality autosomal bi-allelic variants (imputation quality of $R^2 \geq 0.3$, MAF $\geq 1\%$).

For the exome chip content, QC was similarly conducted [18]. In addition, checks specific for exome variants were added [87]. In brief, 96 individuals and 3,818 SNPs were removed, the latter of which had a call rate <0.95 and a Hardy-Weinberg equilibrium p-value <1.0E−05. The final exome chip dataset contains 5,027 participants with 226,233 variants (**S1 Fig**). For the exome chip association analysis, the genotypes were post-processed using zCall with a z-score threshold of six [88]. Genomic positions base on human genome build GRCh37.

### Genome-wide association study of common variants

As previously reported [17,18], GWAS was conducted for GCKD participants with complete genotyping (Omni2.5), eGFR, UACR and log$_2$(OPN) measurement (N = 4,897) data using linear regression of log$_2$(OPN) on SNPs (additive genetic model) with a MAF $\geq 1\%$, adjusted for age, sex, log(eGFR), and log(UACR) (**S1 Fig**). Association analysis was performed using SNPTEST v2.5 [89]. Summary statistics were checked for quality using GWAtoolbox [90] and for inflation using genomic control [91]. A genomic control correction, however, was not requested ($\lambda = 1.01$). Associations with a p-value <5.0E-08 were considered significant. Per chromosome, an index SNP was defined as the SNP with the lowest genome-wide p-value with a 1-Mb interval centered around this SNP. This approach was repeated until no further SNP outside the interval(s) was available passing the genome-wide significance threshold. In order to discover further independent signals, we repeated GWAS analysis for chromosomes with significant results by conditioning on the genotype of the SNP with the lowest association p-value of the respective chromosome. This procedure was repeated until no further genome-wide signal was observed.

Functional annotation of variants was conducted using ANNOVAR[92], SNiPA [93], Open Targets Genetics [94], FAVOR [95], and RegulomeDB [96]. Regional association plots were plotted using LocusZoom v1.3 [97].

### Fine-Mapping

Statistical fine-mapping [21] was carried out as previously described [17] for the two replicated SNPs within a region ±500kb. Approximate Bayes factors (ABFs) were then derived from the original GWAS statistics estimates. The SD prior was chosen as 0.61 because 95% of the effect size estimates fell within the −1.2 to 1.2 interval [21]. The ABF of the SNPs were used to calculate the posterior probability for each variant driving the association signal (PPA, 'causal variant'). Credible sets were determined by summing up PPA-ranked variants until the cumulative PPA was >99%.

## Colocalization analyses

In order to further understand the molecular mechanisms and associated phenotypes underlying the associations, we performed colocalization analyses of the OPN GWAS summary statistics related to the two replicated OPN loci with GWAS summary statistics from three other sources as outlined below. For all colocalization analyses, we used the 'coloc.fast' function from the R package *gtx* with default parameters and prior definitions (https://github.com/tobyjohnson/gtx), an implementation of an adapted version of the colocalization method introduced by Giambartolomei *et al.* [98]. We consider a positive colocalization when the posterior probability of a shared causal variant at the association locus for both traits (H4, p12) was $> 0.8$.

**Gene expression.** First, we used GWAS summary statistics of gene expression data from the GTEx project [99] and the NEPTUNE study [100]. The eQTL data from the GTEx V8 (49 tissues) and the NEPTUNE study (NephQTL from glomerulus and tubulointerstitial kidney portions) were downloaded from the GTEx Portal (https://www.gtexportal.org/home/) and NephQTL web site (http://nephqtl.org/), respectively.

The analysis steps of colocalization have been described in detail elsewhere [17]. Firstly, GWAS summaries of GTEx and NephQTL in genomic regions ±100kb of the two OPN SNPs were extracted. The genes in the extracted GWAS are identified and for each gene, a *cis* window of 500kb flanking the start and end of the gene are defined. Then, for every such *cis* gene window, with at least one SNP having an association p-value $< 0.001$, the GWAS summaries of GTEx and NephQTL tissue as well as the OPN GWAS were extracted and used as input for colocalization analysis.

**Plasma proteome.** In addition, we used GWAS summary statistics of plasma proteins (pGWAS) by Sun *et al.* [22] to run colocalization analysis to identify consistent association signals between OPN and proteins with effects in *cis* as well as in *trans*. In contrast to data from GTEx and NephQTL, the genome-wide available pGWAS summary does allow the assessment of both effects.

In order to detect colocalization with *cis*-pQTLs, pGWAS summary statistics of any protein-gene region (gene region ±500kb, *cis* region) were extracted. Per OPN locus and a 100kb region around it, we checked if any of the *cis* pGWAS extracts overlapped and had a pGWAS association p-value of $<0.05/2$ (Bonferroni correction for two OPN loci). For all hereby selected proteins, we then extracted the protein-gene-region from the OPN GWAS summary statistics and ran colocalization within the protein-gene-region.

For potential colocalization with *trans*-pQTLs, we selected all proteins with pGWAS association p-values $<0.05/2/3,000$ (Bonferroni correction for the two OPN loci and number of proteins evaluated in pGWAS) within a 100kb region around an OPN-associated index SNP. For all hereby selected proteins, colocalization analyses were conducted within the ±500kb region of the OPN-associated index SNP.

For colocalizing proteins, a Gene ontology (GO) enrichment analysis (http://geneontology.org/, [101,102]) in form of a PANTHER [103] overrepresentation test with the two annotation data sets of GO cellular component and GO molecular function (homo sapiens) as references was conducted to assess enriched categories to which identified proteins were assigned to. Overall, 20,595 human genes are mapped to various terms related to cellular component and molecular function. A category is considered enriched if both, the Bonferroni-corrected p-value of the Fisher's exact test and the false discovery rate based on the Benjamini-Hochberg procedure, are $<0.05$.

**UKB diseases.** Finally, we used the GWAS from GeneAtlas database (http://geneatlas.roslin.ed.ac.uk/) to perform colocalization analysis for the two replicated OPN loci (±500kb)

and all UKB binary disease traits that showed genome-wide significant associations (p-value <5.0E-08) in at least one of the two replicated OPN loci. Overall, GeneAtlas comprises GWAS results of 660 binary disease traits of ~450,000 UKB participants [23].

In addition, we adopted the conditional colocalization analysis approach which was firstly applied in a GWAS of plasma proteome [104]. Performing colocalization on conditionally independent association statistics could reveal true colocalization signals that were missing when using marginal association statistics in the presence of multiple independent association signals. We applied GCTA COJO Slct algorithm to identify independent association signals in the OPN region for the seven traits [105], which showed a trait association signal different from the OPN signal (H3: p1.2>0.8). The cleaned and imputed GCKD genotype dataset mentioned before was used as LD reference by GCTA. We set the collinearity cutoff at 0.1 to be conservative. For loci with more than 1 independent signal, an approximate conditional analysis was conducted by GCTA COJO-Cond algorithm to generate conditional association statistics conditioned on the other independent SNPs in the region [105]. Finally the colocalization analyses were performed as before for each of the independent SNPs using the conditional association statistics as input.

### Aggregated rare variant testing

Overall, 4,879 GCKD participants with complete data on genotyping (Exome chip), eGFR, UACR and $\log_2$(OPN) measurements were included in the analysis of aggregated rare variant testing (**S1 Fig**). As previously described [106], two types of rare variant aggregation tests (burden test, sequence kernel association test [SKAT]) implemented in the R package *seqMeta* (v1.6.7, [107]) were conducted using exome chip data and $\log_2$(OPN) measurements (outcome). Per gene, variants with MAF <1% and having a major effect on the gene product (nonsynonymous, stop gain/loss, splicing; "qualifying variants") as annotated by dbNSFP v.2.0 were aggregated [24,25]. Results were filtered to retain genes with cumulative minor allele count (MAC) ≥10 and with ≥2 contributing variants per gene. Analyses were adjusted for age, sex, log(eGFR), and log(UACR). To adjust for multiple testing, the statistical significance level was corrected for the number of assessed genes (N = 17,575) and the two conducted tests: 0.05/(2×17,575) = 1.4E-06. Moreover, analyses were repeated for significantly associated genes additionally adjusted for the two replicated OPN loci.

### Replication of identified loci in Young Finns Study

The three OPN loci identified in the GWAS of GCKD participants were tested for replication in the Cardiovascular Risk in Young Finns Study (YFS) cohort. Here, plasma OPN was measured by enzyme-linked immunosorbent assay (Human Osteopontin Quantikine kit, R&D Systems, USA) from samples thawed for the first time for the assay in 2007. Samples of 2,442 participants and 546,677 genotyped SNPs were available for further analysis after QC and imputation. Further details can be found in **S1 Methods**.

Per selected locus, association analysis of $\log_2$(OPN) on SNP dosage (additive) was performed by fitting linear regression models adjusted for age, sex, and eGFR by using SNPTEST v2.5.4 [89]. GFR was estimated with the MDRD study equation and $\log_2$-transformed prior to analysis [108]. Replication was defined by a one-sided association p-value <0.05/3 (Bonferroni correction for three OPN loci).

## Supporting information

**S1 Fig. Flow chart showing exclusion of patients and analysis sets.**
(PDF)

**S2 Fig. Osteopontin (OPN, ng/mL) measurements in GCKD.**
(PDF)

**S3 Fig. Quantile-Quantile plot of results from GWAS of $\log_2$(OPN).**
(PDF)

**S4 Fig. Manhattan plot of results from GWAS of $\log_2$(OPN).**
(PDF)

**S5 Fig. Regional association plots obtained in the course of conditional analysis to identify independent signals.**
(PDF)

**S6 Fig. Levels of osteopontin ($\log_2$-transformed) in the overall cohort and across genotypes of discovered SNPs in GCKD.**
(PDF)

**S7 Fig. Regional association plot for the region around rs2731673 on chromosome 5.**
(PDF)

**S8 Fig. Effects of rs10011284 and rs4253311 (chromosome 4) on OPN levels.**
(PDF)

**S9 Fig. Relationship of selected GO terms.**
(PDF)

**S10 Fig. Comparing summary statistics for the *KLKB1* locus from OPN GWAS (unconditional statistics, A) with respective results for the UK Biobank phenotype deep venous thrombosis (DVT, conditional statistics, B).**
(PDF)

**S1 Methods. The Cardiovascular Risk in Young Finns Study (YFS) cohort.**
(DOCX)

**S1 Information. List of institutions and investigators participating in the GCKD study.**
(DOCX)

**S1 Table. Genome-wide association results for common variants (MAF$\geq$0.01) with p-value <1E-06.**
(XLSX)

**S2 Table. Extended annotation of the three top SNPs identified in OPN GWAS in the GCKD study.**
(XLSX)

**S3 Table. SNPs in 99% credible sets for the replicated OPN loci.**
(XLSX)

**S4 Table. Colocalization analysis: results for GTEx tissues.**
(XLSX)

**S5 Table. Colocalization analysis: results for pQTLs, in *trans*.**
(XLSX)

**S6 Table. GO overrepresentation analysis results for colocalized pQTLs, in *trans*.**
(XLSX)

**S7 Table. Colocalization analysis: results for GeneAtlas.**
(XLSX)

## Acknowledgments

We are grateful for the willingness of the CKD patients to participate in the GCKD study. The enormous effort of the study personnel of the various regional centers is highly appreciated. We thank the large number of nephrologists who provide routine care for the patients and collaborate with the GCKD study. The GCKD Investigators are listed in the S1 Information. A complete list of nephrologists currently collaborating with the GCKD study is available at (http://gckd.org).

## Author Contributions

**Conceptualization:** Peggy Sekula, Ulla T. Schultheiss.

**Data curation:** Yurong Cheng, Yong Li, Nora Scherer, Franziska Grundner-Culemann, Terho Lehtimäki, Binisha H. Mishra, Olli T. Raitakari, Matthias Nauck, Kai-Uwe Eckardt, Peggy Sekula, Ulla T. Schultheiss.

**Formal analysis:** Yurong Cheng, Yong Li, Nora Scherer, Franziska Grundner-Culemann, Binisha H. Mishra, Peggy Sekula, Ulla T. Schultheiss.

**Funding acquisition:** Terho Lehtimäki, Olli T. Raitakari, Kai-Uwe Eckardt, Peggy Sekula, Ulla T. Schultheiss.

**Investigation:** Binisha H. Mishra, Olli T. Raitakari, Peggy Sekula, Ulla T. Schultheiss.

**Methodology:** Yurong Cheng, Yong Li, Nora Scherer, Franziska Grundner-Culemann, Binisha H. Mishra, Matthias Nauck, Peggy Sekula, Ulla T. Schultheiss.

**Project administration:** Peggy Sekula, Ulla T. Schultheiss.

**Resources:** Peggy Sekula, Ulla T. Schultheiss.

**Software:** Yurong Cheng, Yong Li, Binisha H. Mishra, Peggy Sekula, Ulla T. Schultheiss.

**Supervision:** Nora Scherer, Franziska Grundner-Culemann, Peggy Sekula, Ulla T. Schultheiss.

**Validation:** Yurong Cheng, Yong Li, Nora Scherer, Franziska Grundner-Culemann, Terho Lehtimäki, Binisha H. Mishra, Olli T. Raitakari, Matthias Nauck, Kai-Uwe Eckardt, Peggy Sekula, Ulla T. Schultheiss.

**Visualization:** Yurong Cheng, Yong Li, Peggy Sekula, Ulla T. Schultheiss.

**Writing – original draft:** Peggy Sekula, Ulla T. Schultheiss.

**Writing – review & editing:** Yurong Cheng, Yong Li, Nora Scherer, Franziska Grundner-Culemann, Terho Lehtimäki, Binisha H. Mishra, Olli T. Raitakari, Matthias Nauck, Kai-Uwe Eckardt, Peggy Sekula, Ulla T. Schultheiss.

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
