## [Decision Letter · Decision Letter 0]

2 Dec 2021

Dear Dr Schultheiss,

Thank you very much for submitting your Research Article entitled 'Genetics of osteopontin in patients with chronic kidney disease: the German Chronic Kidney Disease study' to PLOS Genetics.

The manuscript was fully evaluated at the editorial level and by independent peer reviewers. The reviewers appreciated the attention to an important problem, but raised some substantial concerns about the current manuscript. Based on the reviews, we will not be able to accept this version of the manuscript, but we would be willing to review a much-revised version. We cannot, of course, promise publication at that time.

If you decide to revise the manuscript for further consideration at PLOS Genetics, please aim to resubmit within the next 60 days, unless it will take extra time to address the concerns of the reviewers, in which case we would appreciate an expected resubmission date by email to plosgenetics@plos.org.

[LINK]

We are sorry that we cannot be more positive about your manuscript at this stage. Please do not hesitate to contact us if you have any concerns or questions.

Yours sincerely,

Heather J Cordell

Associate Editor

PLOS Genetics

Gregory Barsh

Editor-in-Chief

PLOS Genetics

Reviewer's Responses to Questions

**Comments to the Authors:**

Reviewer #1: The paper of Cheng and colleagues: “Genetics of osteopontin in patients with chronic kidney disease: the German Chronic Kidney Disease study” presents the results of the genome-wide association study of osteopontin in an European chronic kidney disease (CKD) population.

The manuscript perfectly fits in with quite a number of publications related to the role of OPN and OPN-related genes in chronic disorders, including kidney disease.

The Abstract is well written, accurate, intelligible on its own.

The Author Summary is clear and presents the significance of the obtained results objectively.

The Introduction includes the aim of the study and review of the key literature, however, in my opinion it would also be worth quoting the paper of Kaleta et al. (Cells, 2019) - in line 103.

The Results and methods sections are clear. tables and Figures are accurate and accessible. Results and Methods provide all necessary details to support Authors conclusions.

Discussion is well written and presents the main conclusions, compare the obtained results with the avaliable literature. Moreover, the Authors describe the limitations of their study and plans for future research.

To sum up, the value of the article is great. The paper highlights an interesting topic, and is innovative.

The manuscript is written in a good style and easy to read, there are no grammatical, punctuation or linguistic errors. In the manuscript there are good, clear figures and tables that facilitate understanding of the issues described in the text.

The paper is valuable to publish in PLOS Genetics.

Reviewer #2: This is a detailed manuscript describing a GWAS in a ckd patient cohort with available osteopontin measurements. Osteopontin is an enigma and this is the first GWAS to investigate it using GWAS. The results however are limited - a locus near the SPP1 gene encoding osteopontin is identified, together with 2 other loci. In a replication cohort, 2 loci were confirmed.

Some comments and suggestions

Abstract

"OPN levels are known to be associated with adverse kidney outcomes" - this needs qualification...increased levels?

p4

SPP1 gene variants are associated with kidney disease, more detail required here from the 5 refs added

When commenting on the expression patterns of osteopontin (esp in the kidney) important to state which species and how the expression was determined. There are differences between murine rat and human expression. A potential role for osteopontin in vasculature / atherosclerosis should be mentioned and discussed.

Page 5

Levels of OPN increased n av across the eGFR categories - this is a good example of how the text remains unclear or ambiguous. state it increased from x in CKD stage 1 to y in CKD5

Page 7 - some more detail on the relevance of colocalisation studies is required and perhaps a figure to show how the GTEX data integrates with the loci

The rare variant analysis is interesting. Was this performed in other data sets e.g. UKB

Page 11

The ref to osteopontin and nephrolithiasis is from 2007. There is actually very conflicting data surrounding osteopontin and stones - and more update refs and discussion are warranted -eg PMID: 32842990. How does serum osteopontin correlate with urinary osteopontin?

I would like to see more figures within the main body of the manuscript, esp those that enable the functional significance of the loci to be determined. Chromatin annotations are not discussed.

Reviewer #3: Overview:

In “Genetics of osteopontin in patients with chronic kidney disease: the German Chronic Kidney Disease study”, the authors dissect the genetic influences of osteopontin in a large genetic association study. Osteopontin, encoded by SPP1 is implicated with kidney function and the authors hypothesized that a CKD cohort would be ideally suited to identify genetic associations related to the protein. The GWAS of over 3 million genotyped variants (imputed to 7+ million SNPs) yielded three association signals: MEPE-SPP1 (chr 4), KLKB1 (chr 4), and F12-GRK6 (chr 5). The authors tested replication of these signals in the Young Finns Study (a younger, non-CKD cohort) and found evidence of replication for two signals (MEPE-SPP1 and KLKB1). Interestingly, Rare-variant analysis from exome chip SNPs identified a significant variant at a splice-site (rs139555315).

The genetic association methods are well-executed and clearly described within the manuscript.

However, after reviewing, there are several main concerns and several minor suggestions which are listed, below.

Main Concerns:

1)The analysis of OPN in a CKD cohort is a strength of the manuscript, as the authors hypothesize that genetic influences (e.g., SNP associations) may occur within the presence of disease-state (CKD). The authors further support this hypothesis by showing increased OPN levels within greater severity of CKD (e.g, by eGFR category in Figure S2). However, the selected replication cohort (Young Finns Study) is not comparable to the discovery (GCKD) cohort in multiple regards, primarily CKD status (and thus, eGFR). Furthermore, the replication cohort is also different by age and OPN source (plasma versus serum). Since a comparable replication cohort is not available, it seems most appropriate to compare SNP associations across all available OPN-genetics studies (e.g., including those mentioned starting at line 232), perhaps by a summary statistics meta-analysis and/or test of heterogeneity for SNPs in each of the three loci.

2) Would suggest evaluating replication of the locus, not just one index SNP. The associations plots show multiple SNPs in high linkage disequilibrium. The causal SNP is not known (and not necessarily the index SNP), thus, would be most advantageous to test all significantly associated SNPs (for each of the three loci) from the discovery cohort in the replication dataset/s.

3) Should include GCKD plots that depict the log2(OPN) values by genotype for the three index SNPs (maybe as a supplement).

4) While colocalization analysis implicated F11, there is a missed opportunity to highlight that rs4253311 (one of the replicated signals) is an eQTL in multiple tissues for F11 (GTEx V8). So although rs4253311 is intronically located within KLKB1, it might be more appropriate to link it to F11 –which the authors explain F11 is a highly relevant gene for the CKD phenotype.

Minor Concerns and suggestions:

1) Text/tables should indicate human reference build (b37 or b38) for SNP positions.

2) Line 140: rs10011284 appears to be upstream (not downstream?) of SPP1, since SPP1 is on the positive strand (UCSC genome browser, b38)

3) Would recommend designating loci by their gene name/chromosome versus “1st” and “2nd replicated” locus (lines 248 and 299), gene annotations provide better clarity for the reader.

4) For each locus, would recommend including a second association plot that shows SNP signals after adjusting for the index SNP. Given the LD pattern (multiple variants with large p-values and R2 ~0.5 with index SNP) in Figure 1A, am surprised that there was not a secondary signal. A second plot (maybe in supplement) showing the results of the conditional analysis would be helpful.

5) In the table of associations, would denote if the SNPs were directly genotyped or imputed.

6) The aggregate testing method was appropriate but surprised authors did not use SKAT-O which is a unified optimal test of both SKAT and the burden test. –By using the SKAT-O method, it would reduce the number of tests (1 test vs 2 per gene). (This is not a requested revision as it would not impact results, here).

7) Based on the single-variant results (Table 3B), it appears that the splice-variant rs139555314 is likely driving the aggregate test for SPP1. This could be confirmed by running the aggregate test while excluding the splice-variant.

**Have all data underlying the figures and results presented in the manuscript been provided?**

Reviewer #1: Yes

Reviewer #2: Yes

Reviewer #3: Yes

PLOS authors have the option to publish the peer review history of their article (what does this mean?). If published, this will include your full peer review and any attached files.

Reviewer #1: No

Reviewer #2: No

Reviewer #3: No

---

## [Decision Letter · Decision Letter 1]

23 Feb 2022

Dear Dr Schultheiss,

Thank you very much for submitting your Research Article entitled 'Genetics of osteopontin in patients with chronic kidney disease: the German Chronic Kidney Disease study' to PLOS Genetics.

The manuscript was fully evaluated at the editorial level and by independent peer reviewers. The reviewers appreciated the attention to an important topic but identified one remaining minor concern that we ask you address in a revised manuscript.

We therefore ask you to modify the manuscript according to the review recommendations. Your revisions should address the specific points made by each reviewer.

[LINK]

Yours sincerely,

Heather J Cordell

Associate Editor

PLOS Genetics

Gregory Barsh

Editor-in-Chief

PLOS Genetics

Reviewer's Responses to Questions

**Comments to the Authors:**

Reviewer #1: Dear Authors,

In my opinion your corrections increased the value of the manuscript, therefore I believe that the work deserves to be published in the present form.

Reviewer #2: The authors have carefully considered the comments and have reviewed the paper accordingly. The paper is now much more accessible and the findings more clearly displayed.

Reviewer #3: Authors thoroughly addressed previous concerns. The addition of the YFS regional association plots within the main figures is very helpful and nicely supports findings.

One last question is on line 146 "Conditional analysis did not reveal additional independent signals". I assume this means additional signals outside of the three listed regions; however one region (chromosome 4) did exhibit two independent signals (within the same chromosome region). Would suggest better clarifying this in that statement, as identifying two genome-wide significant signals in the same region is very interesting. Biologically this could indicate multiple modes of function regulated by the same locus.

**Have all data underlying the figures and results presented in the manuscript been provided?**

Reviewer #1: Yes

Reviewer #2: Yes

Reviewer #3: Yes

PLOS authors have the option to publish the peer review history of their article (what does this mean?). If published, this will include your full peer review and any attached files.

Reviewer #1: No

Reviewer #2: No

Reviewer #3: No

---

## [Editor Report · Decision Letter 2]

9 Mar 2022

Dear Dr Schultheiss,

We are pleased to inform you that your manuscript entitled "Genetics of osteopontin in patients with chronic kidney disease: the German Chronic Kidney Disease study" has been editorially accepted for publication in PLOS Genetics. Congratulations!

Yours sincerely,

Heather J Cordell

Associate Editor

PLOS Genetics

Gregory Barsh

Editor-in-Chief

PLOS Genetics

Comments from the reviewers (if applicable):

**Data Deposition**

http://datadryad.org/submit?journalID=pgenetics&manu=PGENETICS-D-21-01377R2

**Press Queries**

---

## [Editor Report · Acceptance letter]

1 Apr 2022

PGENETICS-D-21-01377R2 

Genetics of osteopontin in patients with chronic kidney disease: the German Chronic Kidney Disease study 

Dear Dr Schultheiss, 

We are pleased to inform you that your manuscript entitled "Genetics of osteopontin in patients with chronic kidney disease: the German Chronic Kidney Disease study" has been formally accepted for publication in PLOS Genetics! Your manuscript is now with our production department and you will be notified of the publication date in due course.

With kind regards,

Livia Horvath

PLOS Genetics

On behalf of:
